# Full-Length Transcriptome Analysis of Skeletal Muscle of Jiangquan Black Pig at Different Developmental Stages

**DOI:** 10.3390/ijms25116095

**Published:** 2024-05-31

**Authors:** Qi Song, Jinbao Li, Shiyin Li, Hongzhen Cao, Xinlin Jin, Yongqing Zeng, Wei Chen

**Affiliations:** 1Shandong Provincial Key Laboratory of Animal Biotechnology and Disease Control and Prevention, College of Animal Science and Technology, Shandong Agricultural University, Tai’an 271017, China; qisong114@163.com (Q.S.); jinbaol15127735952@163.com (J.L.); garyli1121@163.com (S.L.); hongzhencao@163.com (H.C.); xinlin_jin@163.com (X.J.); yqzeng@sdau.edu.cn (Y.Z.); 2Key Laboratory of Efficient Utilization of Non-Grain Feed Resources (Co-Construction by Ministry and Province), Ministry of Agriculture and Rural Affairs, Tai’an 271017, China

**Keywords:** *Jiangquan black pig*, full-length transcriptome, muscle development, longissimus dorsi muscle

## Abstract

Skeletal muscle grows in response to a combination of genetic and environmental factors, and its growth and development influence the quality of pork. Elucidating the molecular mechanisms regulating the growth and development of skeletal muscle is of great significance to both animal husbandry and farm management. The *Jiangquan black pig* is an excellent pig breed based on the original *Yimeng black pig*, importing the genes of the *Duroc pig* for meat traits, and cultivated through years of scientific selection and breeding. In this study, full-length transcriptome sequencing was performed on three growth stages of *Jiangquan black pigs*, aiming to study the developmental changes in *Jiangquan black pigs* at different developmental stages at the molecular level and to screen the key genes affecting the growth of skeletal muscle in *Jiangquan black pigs*. We performed an enrichment analysis of genes showing differential expression and constructed a protein–protein interaction network with the aim of identifying core genes involved in the development of *Jiangquan black pigs*. Notably, genes such as *TNNI2*, *TMOD4*, *PLDIM3*, *MYOZ1,* and *MYH1* may be potential regulators of muscle development in *Jiangquan black pigs*. Our results contribute to the understanding of the molecular mechanisms of skeletal muscle development in this pig breed, which will facilitate molecular breeding efforts and the development of pig breeds to meet the needs of the livestock industry.

## 1. Introduction

Muscle growth and development are closely related to meat production and are a major factor in the overall growth [1]. The skeletal muscle of pigs is an important source of meat products and an ideal model for muscle growth and development studies [2]. Its growth and development are a complex biological process, involving the interaction of multiple functional genes and with the environment [3]. Skeletal muscle tissue primarily consists of myofibers, which are formed through a process called myogenesis. This process occurs during the prenatal and postnatal stages of growth, development, and regeneration. Myofibers are derived from myoblasts, which undergo proliferation, and fusion to form myotubes, and ultimately differentiate into mature myofibers. Prenatal myogenesis involves primary and secondary myogenesis. Primary myogenesis originates from embryonic myofibroblasts, while secondary myogenesis arises from fetal myofibroblasts. These processes occur at different time points during gestation, with primary myogenesis typically taking place between approximately 35–60 days and secondary myogenesis occurring between 55–90 days [4].

*Jiangquan black pig* is a crossbred based on the *Yimeng black pig* with introgression of *Duroc pig* which accounts for about 35% of the overall genetics. In addition to maintaining the main advantages of the original *Yimeng black pig*, such as physical appearance, high disease resistance, roughage tolerance, and good meat quality, the *Jiangquan black pig* also has the characteristics of fast weight gain and high lean meat rate, which has great economic value.

Different pig breeds exhibit variations in gene expression within their skeletal muscle tissues, which can have an impact on muscle growth and the quality of meat produced [5]. These differentially expressed genes contribute to the distinctive characteristics observed in various pig breeds. The CIDE family of genes regulates lipid metabolism and contributes to fat deposition, which may lead to obesity. The expression of CIDEA and CIDEC genes in adipose and longissimus dorsi muscle tissues was significantly higher in obese pigs than in lean pigs [6]. *Jinhua pigs*, with their high intramuscular fat content, high fertility, and slow growth, are now widely used in ham production [7]. The *Ningxiang pig* has long and slender fibers, with rich and evenly distributed fat between tissues, and tender meat quality [8]. The characteristics of *Yorkshire pigs* are fast growth, large body size, and minimal fat deposition [9]. Muscle development and growth have a direct impact on the quality and yield of meat, making it a critical and economically significant aspect of pig farming [10]. Sequencing transcribed genes from *Jiangquan black pigs* will help us study the growth and development of skeletal muscle in *Jiangquan black pigs*, as gene expression levels vary between different breeds of the same species.

Oxford Nanopore Technologies (ONT) sequencing is the next generation of nanopore-based real-time electrical signal sequencing of single molecules [11]. Unlike short-read technology used in RNA sequencing (RNA-Seq), the long-read segment technology that provides full-length transcripts can accurately identify complex gene structures such as variable splicing and fusion genes. This study utilized ONT sequencing technology to obtain the full-length transcriptome of *Jiangquan black pigs* at gestational days 60 and 75, as well as at postnatal day 240. By studying and analyzing the genes differentially expressed in the longissimus of *Jiangquan black pigs* at 60 days of gestation, 75 days of gestation, and 240 days after birth, our data will help reveal a better understanding the molecular mechanism of skeletal muscle development and meat formation, but also promote the process of molecular breeding and the cultivation of excellent pig breeds to meet the needs of the development of the animal husbandry industry.

## 2. Results

### 2.1. Histomorphologic Changes in the Longissimus Dorsi Muscle

Morphological examination of the *longissimus dorsi* muscles of *Jiangquan black pigs* at 60 days of gestation (E60), 75 days of gestation (E75), and 240 days after birth (A240) revealed that the diameter and area of muscle fibers increased progressively with age (Figure 1A–C). There was a statistically significant difference in the diameter and area of the muscle bundles at the three different time points (Figure 1D,E).

### 2.2. Overview of Full-Length Transcriptome Data

Sequencing yielded an average of 5,828,888 clean sequences of nine samples, with full-length sequences ranging from 4,471,950 reads to 5,316,953 reads for each sample. The ratios of full-length sequences to the number of clean reads were all above 80%. The average length of each sequence was 511 bp, and more than 90% of each sample could be uniquely aligned with the reference genome (Table 1).

### 2.3. Functional Annotation of Novel Transcript

A comparison of the non-redundant transcripts of all samples to the known annotations of the reference genome identified 9374 novel transcripts (Figure 2A). The results for novel transcript type and number are shown in Figure 2B.

The 9374 novel transcripts were annotated revealing 1192 known transcripts in at least one database. The KEGG database annotated a total of 190 transcripts. In the pathway database, there were annotations for 189 transcripts. Annotations for 694 transcripts were found in the Nr database. There are 241 transcripts annotated to the GO database. In the KOG database, 22 transcripts were identified, while in the pfam database, annotations were possible for 252 transcripts. Additionally, 89 transcripts were annotatable in the TF database (Table 2).

Annotation of novel transcripts from the NR database revealed 360 transcripts attributed to sus scrofa, 25 transcripts to muntiacus reevesi, and 22 transcripts to camelus dromedarius (Figure 2C).

### 2.4. Identification of LncRNA

A total of 26 lncRNAs were predicted using three methods (Figure 3A). According to the position of lncRNAs on the annotation information of the reference genome, 15 lncRNAs were identified as long spacer non-coding RNAs, five lncRNAs as antisense lncRNAs, one lncRNA as intronic lncRNA, and five lncRNAs as positive-sense lncRNAs (Figure 3B). The prediction yielded 109 cis-acting target genes and 5,343 trans-acting target genes. Partial lncRNAs and their regulated target genes are depicted in Figure 3C. Among the cis-acting target genes, two genes (*TMOD4* and *PDLIM3*) were differentially expressed among all three groups. These were the target genes of two lncRNAs, TMOD4.t1 and PDLIM3.t5. Among the trans-acting target genes, 96 genes were differentially expressed among all three groups. Enrichment analysis of the 96 differentially expressed genes yielded a total of 90 biological process (BP) terms, 18 cellular component (CC) terms, and 3 molecular function (MF) terms. BP terms were primarily enriched in processes such as muscle fiber assembly and muscle system processes, while CC terms were mainly enriched in terms related to muscle fibers and contractile muscle fibers. MF terms were enriched in terms related to actin binding and actin filament binding (Figure 3D–G). 

*TMOD4* and *PDLIM3* genes are regulated by TMOD4.t1 and PDLIM3.t5, respectively, and are enriched for terms such as muscle structure development, so it is hypothesized that TMOD4.t1 and PDLIM3.t5 are able to regulate skeletal muscle growth and development. 

### 2.5. SSR Analysis

The combined sequence length of all SSRs amounted to 275,294,913 bp. Among the 36,601 sequences identified, 91,376 SSRs were detected (Table 3). Seven types of SSRs were identified: single nucleotide, dinucleotide, trinucleotide, tetranucleotide, pentanucleotide, hexanucleotide, and compound SSRs (Figure 4).

### 2.6. Alternative Splicing

Alternative splicing (AS) is an important mechanism in the regulation of gene expression. A total of 12 differentially expressed AS events were detected in this study. In this study, exon skipping (SE) and alternative last exon (AL) are the two most common AS events (Table 4).

### 2.7. Functional Annotation and Enrichment Analysis of Differentially Expressed Genes

Between the gestational age of 60 and 75 days, we identified 1201 differentially expressed genes, with 434 upregulated and 767 downregulated. Between 60 days of gestational age and 240 days after birth, the number of differentially expressed genes increased to 4246, of which 2262 were upregulated and 1984 were downregulated. Finally, a total of 3825 differentially expressed genes were obtained between 240 days postnatal and 75 days postnatal, of which 2002 were upregulated and 1823 were downregulated (Figure 5A–C). There were 234 genes that were differentially expressed among all three groups (Figure 5D).

For all identified differentially expressed genes, the relationship between samples and genes was hierarchically clustered according to their gene expression levels, and the clustering results were presented using a heat map (Figure 5E).

The 234 differentially expressed genes underwent enrichment analysis. BP annotation yielded 185 terms, predominantly enriched in processes such as myofibril assembly, morphogenesis of cellular components, skeletal muscle contraction, muscular systemic processes, and muscle architecture development. CC annotation produced 24 terms, mainly enriched in components like myofibrils, contractile fibroblasts, myonectoderms, and the actin skeleton. MF annotation yielded 12 terms, primarily enriched in activities such as cellular backbone protein binding, actin binding, glycosaminoglycan binding protein, and others (Figure 5F–H).

Among the KEGG pathways enriched, the major pathways enriched include amino acid biosynthesis, cGMP-PKG signaling pathway, glycolysis and glucose metabolism, and calcium signaling pathway (Figure 5I). 

### 2.8. Time Series Expression Pattern Clustering

In order to investigate the dynamic changes in the longissimus dorsi muscle differentially expressed genes in different developmental stages of *Jiangquan black pigs*, the expression patterns of the 234 differentially expressed genes screened were clustered and analyzed using STEM (v 1.3.12) software, and a total of 16 differentially expressed genes expression patterns were obtained, of which three were significantly enriched during the development of the longissimus dorsi muscle of the *Jiangquan black pig* (*p* < 0.05), and Profile 0 and 3 showed a similar pattern of expression. The differentially expressed genes showed a downregulation trend during development, while Profile 8 showed an upregulation trend (Figure 6).

### 2.9. DEG Protein–Protein Interaction Analysis and Hub Gene Identification

To further explore the interactions between differential genes, a new interaction network was constructed using Cytoscape software (v3.10.0). Network analysis provides a rapid and visual understanding of the complex gene regulatory network of skeletal muscle development. A protein–protein interaction network was constructed (Figure 7). Among them, nodes with the largest number of neighbors were considered as hub genes, with *TNNI2*, *MYH1*, *TNNC2*, *ACTA1,* and *MYOZ1* being the most critical node genes.

### 2.10. Validation of the Results by qRT-PCR

The accuracy of the sequencing data was verified by qRT-PCR validation results for seven randomly selected genes (PPP3R1, B2M, TNNC2, TPM1, TMOD4, ENO3, and BTG2) from 234 differentially expressed genes. The experimental results of these genes were consistent with the sequencing results, indicating that the sequencing results in this study were reliable (Figure 8).

## 3. Discussion

Meat quality is a critical economic trait of pigs, which, in turn, is determined by the features of muscle fibers. Skeletal muscle quantity and quality are considered to be the main indicators of meat quality [12]. Skeletal muscle growth and development is a rather complex process, including the differentiation of muscle-derived stem cells into muscle cells and monocytes, cell fusion into multinucleated myotubes, and maturation of myofibers, which is regulated by numerous transcription factors and signaling pathways [13].

*Jiangquan black pig* is a cross between *Yimeng black pig* sows and *Duroc boars*, which produces hybrid F1 generation, selects F1 generation sows and backcrosses with *Yimeng black pig boars*, and cultivates a good pig breed after several generations of crossbreeding and stereotyping. This breed is touted for its exceptional meat quality and resistance to roughage. Despite its favorable traits, research on the excellent attributes of the *Jiangquan black pig* remains scarce. To investigate the molecular mechanisms behind skeletal muscle development in *Jiangquan black pigs*, we conducted long-read transcriptome sequencing of their longissimus dorsi muscle. The findings of this study possess the potential to enhance the quality of the *Jiangquan black pig* genetically and on a molecular basis.

Muscle development in the porcine embryo is divided into two stages, the first occurring on embryonic days 35–60 and primarily involving the formation of primary fibers, and the second occurring on embryonic days 54–90 and primarily involving the formation of secondary fibers [1]. After birth, the number of myofibers no longer changes, and muscle growth is mainly based on the increase in myofiber size.

During the process of skeletal muscle development, a sequence of crucial genes is expressed in an organized and regulated manner, ultimately resulting in the formation of skeletal muscle tissue. The myogenic regulatory factor families (*MRFs*) and myocyte enhancer factor family (*MEF2*) are significant contributors to muscle development. The *MyoD* gene was identified as the inaugural specialized factor that facilitated the transformation of non-skeletal muscle cells into skeletal muscle cells. This gene is a component of the *MRFs* family [14]. Myogenic differentiation is reliant on *MyoD*, which interacts with *MEF2* family members’ proteins to create a synergistic effect. The *MyoD* gene’s proteins are vital in the formation of skeletal muscles [15]. *MEF2C* is a transcription factor that has regulatory effects on cardiovascular growth and skeletal muscle development. It plays a significant role in regulating gene expression in various tissues [16].

### 3.1. Differentially Expressed Genes STEM Cluster Analysis

In this study, cluster analysis of differentially expressed genes expression patterns at three stages during skeletal muscle muscular development of the *Jiangquan black pig* was performed using STEM (v1.3.11) [17] software, and a total of 16 differentially expressed gene expression patterns were obtained, of which 3 were significantly enriched during the development of the longissimus dorsi muscle of the *Jiangquan black pig* (*p* < 0.05), and Profile zero and Profile three showed similar expression patterns, and differentially expressed genes showed a trend of downregulation during the development of the longissimus dorsi muscle of the *Jiangquan black pig*, whereas genotypes 8 showed an upregulation trend.

### 3.2. Differentially Expressed Genes Analysis

GO enrichment results indicated that *TMOD4*, *CASQ1*, *PDLIM3*, *MYOZ1*, *ACTA1* and other genes were involved in the development of muscle structure, which might be closely related to the development of the longissimus dorsi muscle of *Jiangquan black pigs*. It was shown that *TMOD4* plays an important role in filament length regulation and myogenic fiber assembly [18]. The *TMOD4* gene can act as a switch between muscle growth and adipogenesis, leading to balanced development between skeletal muscle and fat [19]. *TMOD4* is the major *TMOD* isoform in mammalian skeletal muscle and has a role in cardiac muscle comparable to that of *TMOD1* [20]. *TMOD4* gene has nine exons and eight introns, and there are two binding sites in the upstream sequence of its promoter, which are related to the growth of muscle fibers and the production of adipose molecules, respectively. When *TMOD4* is expressed at a high level, it promotes the deposition of adipose and inhibits the development of muscle fibers; and when it is expressed at a low level, it inhibits adipose and promotes the growth of muscle fibers [18]. *PDLIM3* encodes the PDLIM3 protein from the LIM family and contains the PDZ and LIM structural domains [21]. The *PDLIM3* gene is highly expressed in skeletal muscle and PDLIM3 plays a major role in maintaining the differentiation of C2C12 myoblasts [22]. The *PDLIM3* gene has important functions in the maintenance of myoblast stability, differentiation, normal muscle development, cell signaling, cell proliferation, and integration of cytoskeletal structures [23]. A novel *PDLIM3* splice variant was identified in porcine skeletal muscle that was expressed only in heart and skeletal muscle, with the highest expression in adult pig skeletal muscle [24].

*CASQ1* is the most important Ca^2+^ binding protein, which is located in the sarcoplasmic reticulum of skeletal and cardiac muscle [25]. In skeletal muscle, the first role of *CASQ1* is to buffer Ca^2+^ in the sarcoplasmic reticulum with a low affinity and high capacity for Ca^2+^ binding in order to allow for rapid Ca^2+^ release during muscle contraction. The second role of *CASQ1* in skeletal muscle is to act as a Ca^2+^ sensor in the sarcoplasmic reticulum. *CASQ1* acts as a Ca^2+^ sensor in the sarcoplasmic reticulum by sensing Ca^2+^ in the sarcoplasmic reticulum and regulating, in a conformation-dependent manner, the release of Ca^2+^ release from the sarcoplasmic reticulum into the cytoplasm via RyR1 [26].

In swine, *Myoz1* expression is mainly observed in the heart, skeletal muscle, and smooth muscle tissues [27]. The expression of *Myoz1* showed a gradual increase in porcine skeletal muscle development from embryonic day 33 to adulthood. However, it did not participate in cell proliferation during C2C12 cell differentiation [28]. *Myoz1* is a potential candidate gene for influencing meat quality traits in livestock and poultry due to its irreplaceable role in signal transduction and muscle fiber type differentiation. The expression of *Myoz1* was gradually upregulated with the maturation of myofibers during muscle regeneration, and the re-expression of *Myoz1* protein was closely related to the degree of myofiber regeneration [29]. It was shown that skeletal muscle alpha-actinin (*ACTA1*), encoded by the *ACTA1* gene, belongs to the actin family, which consists of 13 isoforms in humans [30]. Co-expressed with cardiac alpha-actin in adult skeletal muscle tissues, *ACTA1* plays an important role in cell contraction, movement, and structural and morphological changes as a major component of skeletal muscle [31]. *ACTA1* is extensively involved in the assembly of muscle filaments, the development of skeletal muscle fibers, and the movement of cells and organelles. The *TNNC2* gene encodes the fast skeletal muscle C subunit of the troponin complex, which plays a key role in regulating striated muscle contraction and may be one of the candidate genes for pork quality. The expression of this gene is closely related to the quality characteristics of pork, so its study will help us to understand the mechanism of pork quality formation [32].

### 3.3. Analysis of GO and KEGG Pathway

Skeletal muscle development is regulated by various signaling pathways, such as the Wnt signaling pathway, the Notch signaling pathway, and the mTOR signaling pathway [33]. The KEGG pathway is mainly enriched to the amino acid biosynthesis pathway and cGMP-PKG signaling pathway, among others. Leucine is an important functional amino acid [34], which is able to stimulate protein synthesis in the skeletal muscle of newborn piglets through the mTOR signaling pathway [35]. The cGMP-PKG signaling pathway is implicated in the regulation of glucose uptake in skeletal muscle [36].

In this study, GO was enriched to terms such as myofibril assembly and skeletal muscle contraction.

### 3.4. AS Events in Skeletal Muscle Developmental Process

AS is a meticulously regulated biological process pivotal in generating extensive transcriptomic and proteomic diversity, significantly enhancing the complexity of eukaryotic organisms [37]. Research indicates that 40% of protein modifications stem from alternative splicing, a process characterized by the generation of multiple mRNA products from a single precursor mRNA. These mRNA products can be translated into proteins with diverse structures or functions [38]. Alternative splicing plays an important role in the growth and development of skeletal muscle [39]. In this study, it was observed that SE (skipped exon) events and AL (alternative last exon) events constituted the majority of differentially expressed AS events, indicating their prevalence in the process of skeletal muscle development in *Jiangquan black pigs*.

### 3.5. Protein Interaction Network Analysis

The PPI interaction network showed that the *TNNI2* gene is a core gene. *TNNI2* is primarily expressed in muscle cells and functions to regulate muscle contraction as part of the troponin complex, which binds to calcium ions [40]. It has been shown that the expression of the *TNNI2* gene gradually increases with sarcomere regeneration, allowing the assessment of myofiber maturity [30].

## 4. Conclusions

In this study, the full-length transcriptome of the longissimus dorsi muscle of was obtained using ONT sequencing technology, and 9374 novel transcripts were obtained. Based on GO enrichment analysis, a large number of genes related to skeletal muscle growth and development were obtained, and we hypothesized that *TNNI2*, *MYOZ1*, *PDLIM3*, *TMOD4,* and *ACTA1* play key roles in the skeletal muscle growth and development of Jiangquan black pigs.

## 5. Materials and Methods

### 5.1. Ethics Statement and Sample Collection

The animal study was reviewed and approved by the Animal Ethics Committee of Shandong Agricultural University and performed in accordance with the Committee’s guidelines and regulations (No. 2004006).

*Jiangquan black pigs* were obtained from Shandong Linyi Jiangquan Agricultural and Animal Husbandry Co., Ltd. During the prenatal phase of the sow’s life, three sows were slaughtered at 60 and 75 days after mating and the longissimus dorsi muscle tissue was dissected from the embryos, and three 240-day-old sows were slaughtered, and the longissimus dorsi muscle tissue was dissected. Three 240-day-old sows were fed from birth under the same management and conditions and with the same nutritional standards. After slaughter, the longissimus dorsi muscle was collected from the same part of each pig, immediately frozen in liquid nitrogen (−196 °C), and then transferred to a refrigerator at −80 °C for storage pending RNA extraction.

### 5.2. Hematoxylin and Eosin (HE) Staining

The longissimus dorsi muscle tissues of *Jiangquan black pigs* were taken and fixed in 4% formaldehyde solution for more than 24 h. The fixed tissues were dehydrated in an Excelsior AS dehydrator, then embedded using a HistoSTAR embedding machine, and then the waxed tissue blocks were cut into five μm thick sections using a Microm HM 355S slicer. They were stained using hematoxylin and eosin and finally observed using a microscope, and the diameter and area of muscle fibers were counted using Image Pro Plus 6.0.

### 5.3. RNA Extraction and Sequencing Analysis

The RNA extraction kit (TIANGEN, Beijing, China) was used to extract RNA from the longissimus dorsi of *Jiangquan black pigs*. The extracted RNA was analyzed by NanoDrop One spectrophotometer (NanoDrop Technologies, Wilmington, DE, USA) and Qubit 3.0 Fluorometer (Life Technologies, Carlsbad, CA, USA) for quality testing.

The cDNA libraries were prepared using cDNA-PCR sequencing kits (SQK-PCS109 and SQK-PBK004) provided by ONT. The prepared cDNA libraries were added the libraries into the R9.4 sequencing chip, and PromethION sequencer (Oxford Nanopore Technologies, Oxford, UK) was on-boarded for sequencing (Wuhan Benagen Technology Co., Ltd.) for 48~72 h. The obtained sequencing data were subjected to quality control.

Filter the raw fastq data using NanoFilt (v2.8.0) [41] to obtain valid data for subsequent analysis, excluding sequences with quality scores less than 7 and sequences shorter than 50 bp.

Use Pychopper (v2.4.0) software to identify full-length sequences in valid sequencing data and use minimap2 (v2.17) [42] to compare the filtered full-length sequences with the reference genome (Sus Scrofa 11.1).

### 5.4. Identification of lncRNA

Three computational methods CNCI (v 2.0) [43], CPC2 (v1.0.1) [44] software, and pfam [45] were used for the prediction of lncRNA in novel transcripts. LncRNA mainly functions by regulating the expression of target genes. The basic principle of cis-acting target gene prediction is that the function of lncRNAs is related to the genes close to their coordinates, and we screen out the genes in the proximity of lncRNAs as its target genes. The basic principle of trans-acting target gene prediction is that the function of lncRNA does not depend on the position of the target genes but is related to the genes it co-expresses. The Pearson correlation coefficient method was used to analyze the correlation between lncRNAs and genes among the samples, and the genes whose correlation coefficients were greater than 0.9 were selected as their target genes.

### 5.5. SSR Analysis

SSR (simple sequence repeats), also known as short tandem repeats or microsatellite markers. It is a class of tens of nucleotides long repeat sequences consisting of a few nucleotides (1–6) as repeat units, which are relatively short in length and widely and uniformly distributed in the genomes of eukaryotic organisms. Simple sequence repeats of the transcriptome were identified using MISA (v1.0) [46].

### 5.6. Alternative Splicing

Gene transcription generates precursor mRNAs (pre-mRNAs) that are spliced in a variety of ways, selecting different exons to produce different mature mRNAs, which are translated into different proteins that constitute the diversity of biological traits. The variable splicing types present in each sample were obtained using suppa2 (v2.3) [47] software.

### 5.7. Differential Expression Analysis and Gene Functional Annotation

Performing differentially expressed genes analysis using DESeq2 (v1.26.0) [48], significance filtering is conducted based on the False Discovery Rate (FDR), with the criteria of *p*-value less than 0.05 and |log_2_FoldChange| greater than 1.

Gene Ontology (GO) (http://geneontology.org/) (accessed: 16 June 2023) enrichment analysis of differentially expressed genes was conducted through the GO database to find the main biological functions. The pathway enrichment analysis of differentially expressed genes was conducted through the KEGG database (https://github.com/xmao/kobas) (accessed: 16 June 2023) to elucidate the major pathways involved in muscle growth and development.

### 5.8. Time Series Expression Pattern Clustering

To explore the gene regulation pattern of muscle growth and development in Jiangquan black pigs, the expression levels of differentially expressed genes with similar expression patterns were clustered and analyzed using STEM (1.3.13) [49] software.

### 5.9. Protein–Protein Interaction (PPI)

The 234 differentially expressed genes were compared with proteins in the STRING database (http://string-db.org/) (accessed: 10 October 2023) to find homologous proteins and construct an interaction network based on the interaction relationships of the homologous proteins, and then visualize the PPIs of these DEGs in Cytoscape (v3.10.0).

### 5.10. Quantitative Real-Time PCR (qRT-PCR) Validation

Among the differentially expressed genes among the three groups, six differentially expressed genes (*PPP3R1*, *B2M*, *TNNC2*, *TPM1*, *ENO3,* and *BTG2*) were randomly selected for fluorescence quantification and validation. Primers were designed using Primer Premier 5.0 software, and the primer sequences are shown in Table 5. The primers were synthesized by Bioengineering Co., (Shanghai, China).

Total RNA was extracted from the longissimus dorsi muscles of *Jiangquan black pigs* at different growth stages and reverse transcribed into cDNA using Evo M-MLV reverse transcription kit (Accurate Biotechnology, ChangSha, China). cDNA was then analyzed by using SYBR Green Pro Taq HS premixed qPCR kit (Accurate BIotechnology, ChangSha, China) for qPCR. The reaction system (20 μL) consisted of 10 μL of 2X SYBR Green Pro Taq HS Premix*6, 0.4 μL of upstream primer, 0.4 μL of downstream primer, 0.4 μL of cDNA, and 7.2 μL of ddH_2_O. The PCR reaction conditions were as follows: pre-denaturation at pre-denaturation at 95 °C for 30 s, denaturation at 95 °C for 5 s, and annealing at 60 °C for 30 s in 40 cycles. Relative gene expression levels were calculated using the 2^−ΔΔCt^ method.

### 5.11. Statistical Analysis

Significance analysis was performed using IBM SPSS Statistics 22. The unpaired Student *t*-test (two-tailed) was used for the comparison between two unpaired groups. One-way analysis of variance (ANOVA) was applied for multi-group data comparison. The criterion for the significance of difference was *p* < 0.05. Data visualization was performed using GraphPad Prism 9.5.1.

## Figures and Tables

**Figure 1 ijms-25-06095-f001:**
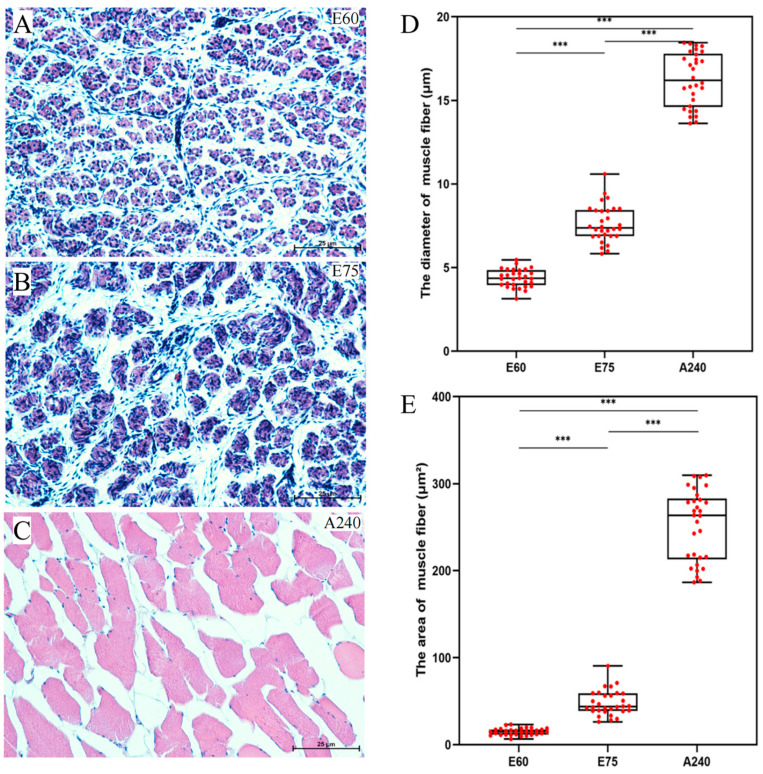
Phenotypic characterization of skeletal muscle in *Jiangquan black pigs*. (**A**–**C**) Morphological changes in longissimus dorsi muscle tissues (200×). (**D**) Changes in muscle fiber diameter over time. (**E**) Changes in myofiber area in different periods. *** represents highly significant differences (*p* < 0.001).

**Figure 2 ijms-25-06095-f002:**
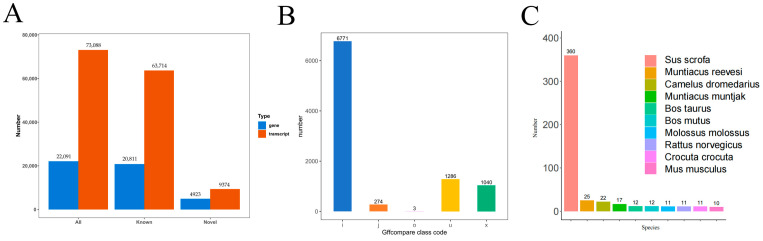
Analysis of novel transcripts. (**A**) Statistics on the number of novel transcripts and new genes. (**B**) Plot of the number of novel transcript types in the samples. “o” denotes regions on the same strand overlapping with reference exons, “j” signifies at least one matching multi-exon, “x” represents exon overlap on the opposite strand, “i” indicates introns completely contained within the reference transcript, and “u” denotes unknown new transcripts. (**C**) Nr annotation result statistics.

**Figure 3 ijms-25-06095-f003:**
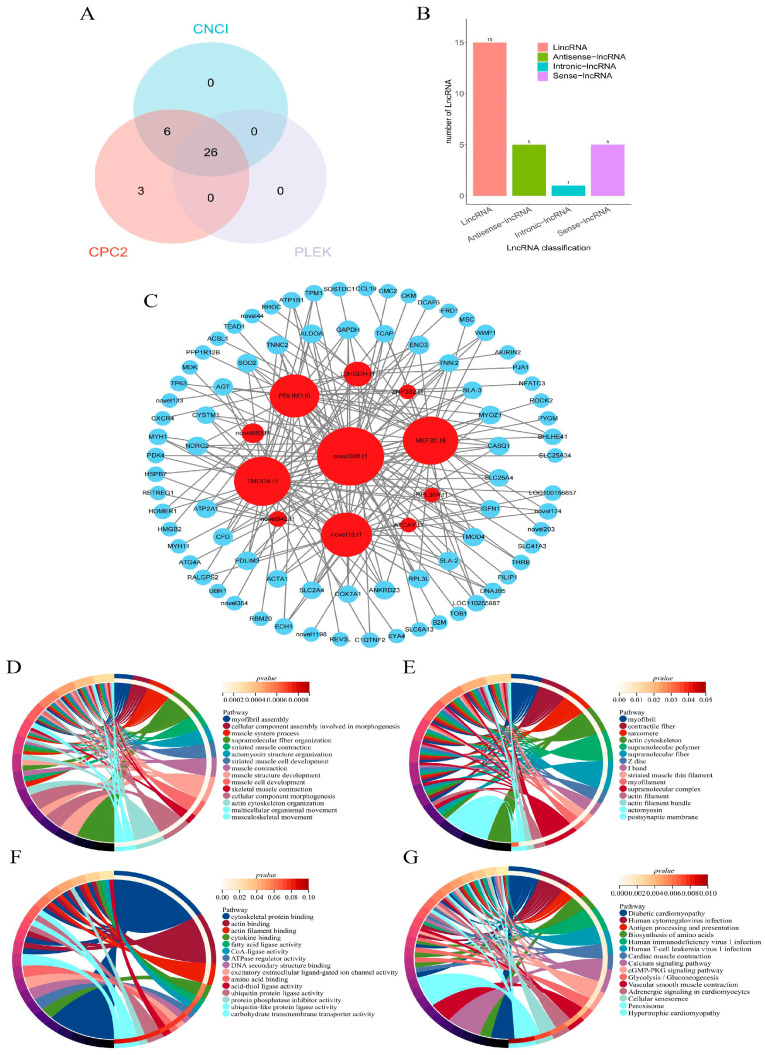
LncRNA quantity counting and target gene enrichment analysis. LncRNA analysis. (**A**) Three methods to predict lncRNA quantity Venn diagram. (**B**) LncRNA quantity statistics. (**C**) Partial lncRNAs with their regulated target genes, lncRNAs in red, and their regulated target genes in blue. (**D**) Target gene GO enrichment analysis. Enrichment results of target gene biological processes. (**E**) Enrichment results for cellular components of target genes. (**F**) Enrichment results for target gene molecular function. (**G**) Target gene KEGG enrichment results.

**Figure 4 ijms-25-06095-f004:**
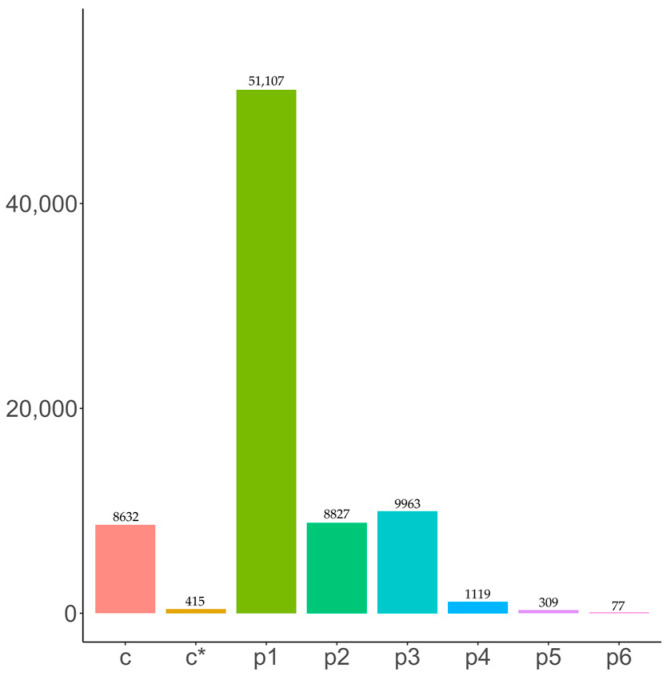
SSR number statistics and type. c: complex repeat type, c*: complex repeat type, p1: single base repeat, p2: two base repeat, p3: three base repeat, p4: four base repeat, p5: five base repeats, p6: six base repeats.

**Figure 5 ijms-25-06095-f005:**
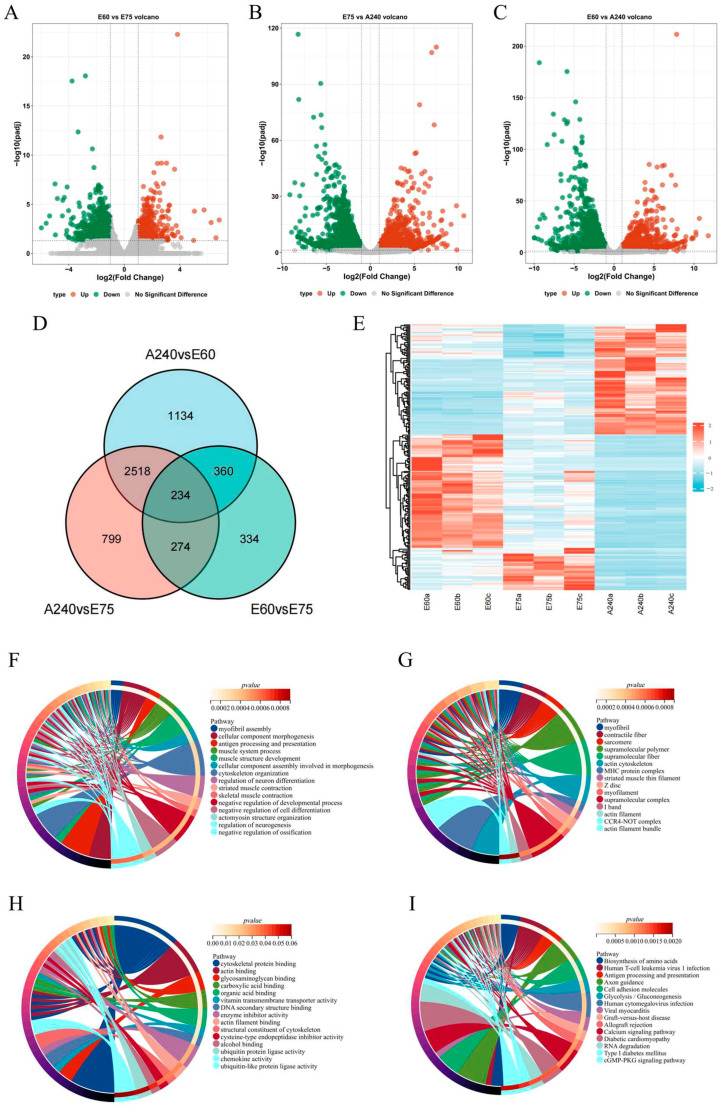
Analysis of differentially expressed genes and GO enrichment analysis of differentially expressed genes. (**A**): Volcano plots of differentially expressed genes at 60 days of gestational age and 75 days of gestational age. (**B**): Volcano plot of differentially expressed genes at 75 days of gestational age and 240 days after birth. (**C**): Volcano plot of differentially expressed genes at 60 days of gestational age and 240 days after birth. (**D**): differentially expressed genes in three developmental stages. (**E**): Heatmap displaying gene expression patterns. (**F**): Enrichment results of differentially expressed gene biological processes. (**G**): Enrichment results for cellular components of differentially expressed genes. (**H**): Enrichment results for differentially expressed gene molecular function. (**I**): Enrichment results of differentially expressed gene KEGG.

**Figure 6 ijms-25-06095-f006:**
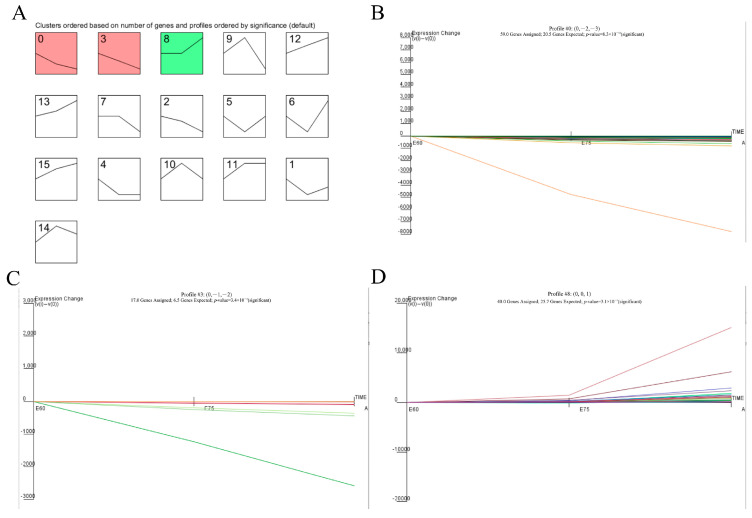
STEM clustering analysis of differentially expressed genes in different developmental stages of *Jiangquan black pigs*. (**A**): A total of 16 DEG patterns were obtained at different developmental stages of *Jiangquan black pigs*. (**B**–**D**): The expression patterns of three DEGs were significantly enriched in *Jiangquan black pigs*.

**Figure 7 ijms-25-06095-f007:**
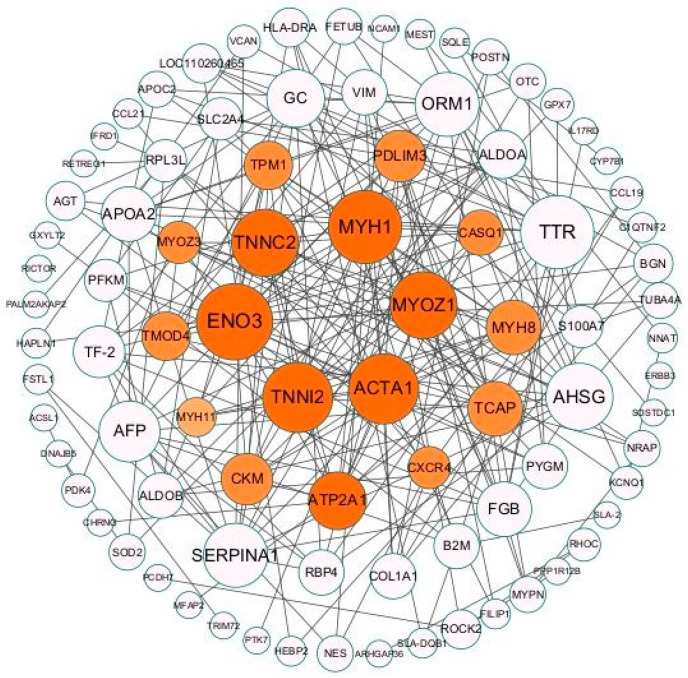
Protein–protein interaction networks. Genes associated with skeletal muscle development are shown in orange.

**Figure 8 ijms-25-06095-f008:**
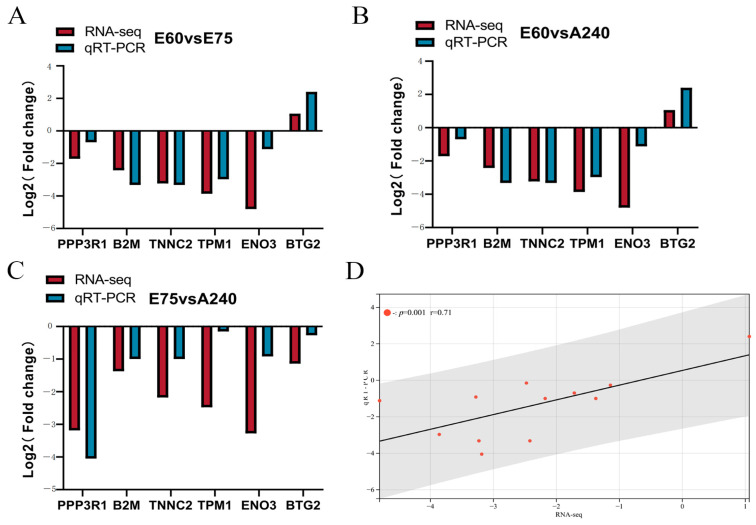
Sequencing data validation. (**A**) Comparison of sequencing data and qRT-PCR results for E60 and E75. (**B**) Comparison of sequencing data and qRT-PCR results for E75 and A240. (**C**) Comparison of sequencing data and qRT-PCR results for E60 and A240. (**D**) Correlation of qRT-PCR and sequencing data.

**Table 1 ijms-25-06095-t001:** Number of full-length reads and full-length reads mapped on genome.

Sample Name	Number of Clean Reads	Number of Full-Length Reads	Full-Length Percentage (FL%)	Mapped Reads	Mapped Rates %
E60a	6,528,793	5,317,523	81.45	4,847,341	91.16
E60b	5,829,456	4,676,769	80.23	4,317,419	92.32
E60c	5,723,474	4,637,041	81.02	4,326,748	93.31
E75a	5,477,552	4,471,950	81.64	4,212,440	94.2
E75b	5,606,283	4,567,562	81.44	4,256,781	93.2
E75c	5,820,493	4,734,003	81.33	4,369,074	92.29
A240a	5,569,209	4,527,703	81.3	4,265,383	94.21
A240b	6,456,405	5,316,953	82.35	4,915,944	92.46
A240c	5,724,512	4,685,954	81.86	4,398,165	93.86

**Table 2 ijms-25-06095-t002:** Novel transcript annotation information.

Item	Count	Percentage (%)
All	9374	100.00
Annotation	1192	12.72
KEGG	190	2.03
Pathway	189	2.02
Nr	694	7.40
Uniprot	1175	12.53
GO	241	2.57
KOG	22	0.23
Pfam	252	2.69
TF	89	0.95

**Table 3 ijms-25-06095-t003:** Type and number of SSRs identified.

Item	Number
Total length of sequence examined (bp)	275,294,913
Total number of SSR	91,376
Total length of SSR	138,122
Relative abundance (SSR/Mb)	331.92
Relative density (bp/Mb)	501.72
Number of SSR containing sequences	36,601
Number of sequences containing more than 1 SSR	19,805
Number of SSRs present in compound formation	8632

**Table 4 ijms-25-06095-t004:** Type and quantity of alternative splicing.

Types of Alternative Splicing	Differential Variable Splicing Quantity	Differential Variable Splicing Ratio (%)
Exon skipping (SE)	6	50
Alternative last exon (AL)	3	25
Alternative first exon (AF)	1	8.33
Alternative 5′ splice site (A5)	1	8.33
Alternative 3′ splice site (A3)	1	8.33

**Table 5 ijms-25-06095-t005:** Primers for real-time quantitative PCR.

Genes	Sequence (5′-3′)
*PPP3R1*	F: TGAGTTACAACAAAATCCC R: ACCATCATCTTCAACACCT
*TNNC2*	F: GCCAGACACCCACCAAAGA R: GAAGATGCGGAAGCACTCA
*ENO3*	F: GGTCCCACTCTACCGTCAC R: CCAGGGCTTCATTGTTCTC
*B2M*	F: TCAGGTTTACTCACGCCAC R: ATCTTCTCCCCGTTTTTCA
*BTG2*	F: AGCACTACAAACACCACTG R: GATGCGATAGGACACTTCA
*TPM1*	F: TGAGTTACAACAAAATCCC R: ACCATCATCTTCAACACCT
*GADPH*	F: ACAGTCAAGGCGGAGAACG R: CGGCAGAAGGGGCAGAGAT

## Data Availability

Data are not publicly available due to ethical reasons. Further enquiries should be directed to the corresponding author.

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
