# Peer review of "Full-Length Transcriptome Analysis of Skeletal Muscle of Jiangquan Black Pig at Different Developmental Stages"

_ijms, 2024, doi:10.3390/ijms25116095_

Round 1

Reviewer 1 Report

Comments and Suggestions for Authors

The study provides a detailed report of the transcriptome of the muscles of pigs.

The study is interesting, but some points have been noted below for correction of the final manuscript.

1.      Please define clearly the objectives of the study.

2.      Please present in detail the gap in the literature that will be filled through the publication of this manuscript.

3.      Please explain what is the advantage of this manuscript over other similar studies already published before.

4.      Methodologies – controls. Please add a subsection in M&M to describe clearly the controls that were used in this study: control materials, control animals, controls procedures.

5.      Methodologies – controls. Please explain why you did not use samples from other breeds of pigs as controls.

6.      Methodologies – analysis. Please describe the statistical techniques employed for each type of data, a general statement is not adequate.

7.      Visualization – figures. Excellent work.

8.      Visualization – tables. Please increase the presentation of results by means of tables and reduce the in-text descriptions.

9.      Discussion. Please divide in two sub-sections for easier flow of reading.

10.  References. There are one or two recent (February and March 2024) references that can be cited as they are relevant with this work and should be discussed versus the current results.

11.  Conclusion. In line with the findings of the study.

Overall. Revision and re-evaluation after submission of the improved version.

Author Response

Dear Reviewer,

Thank you very much for taking the time to review this manuscript. I really appreciate all your comments and suggestions! We use this feedback to improve the quality of the manuscript. The comments below are in italics. Our responses are in normal font and changes/additions to the manuscript are in highlighted text.

Thanks again!

  1. Please define clearly the objectives of the study.

Thanks to your suggestions, we have redefined our research goal, the Jiangquan Black Pig. Jiangquan black pig is a crossbred based on Yimeng black pig with introgression of Duroc pig, of which the Duroc meat trait genes account for about 35%. 

  1. Please present in detail the gap in the literature that will be filled through the publication of this manuscript.

Thank you for your suggestion. In this study, we sequenced the full-length transcriptome of the longissimus dorsi muscle of the Jiangquan black pig at three growth stages, and screened out genes that may be related to the growth of the skeletal muscle of the pig, which can help to understand the mechanism of the development of the skeletal muscle in the Jiangquan black pig.

  1. Please explain what is the advantage of this manuscript over other similar studies already published before.

In this study, full-length transcriptome sequencing was used to accurately identify complex gene structures such as variable splicing and fusion genes.

  1. Methodologies – controls. Please add a subsection in M&M to describe clearly the controls that were used in this study: control materials, control animals, controls procedures.

Thank you for your suggestion, in this study, experiments were conducted using fetuses at 60 days of gestational age, fetuses at 75 days of gestational age and sows at 240 days after birth, and all controls are shown in the manuscript.

  1. Methodologies – controls. Please explain why you did not use samples from other breeds of pigs as controls.

Thank you for your suggestion, as you said, using other pig breeds for control will make our study more complete, so we are considering to compare the transcriptome data of Jiangquan black pigs with other pig breeds in our next study, which will be more helpful for us to elucidate the molecular mechanism of skeletal muscle growth in Jiangquan black pigs.

  1. Methodologies – analysis. Please describe the statistical techniques employed for each type of data, a general statement is not adequate.

Thank you for your suggestion, and as you suggested, we have added this section to the manuscript.

  1. Visualization – figures. Excellent work.

Thank you for the compliment.

  1. Visualization – tables. Please increase the presentation of results by means of tables and reduce the in-text descriptions.

Thanks to your suggestion, we have modified some of the results of Alternative Splicing into a table

  1. Please divide in two sub-sections for easier flow of reading.

Thanks to your suggestion, we have divided the discussion section into subsections based on content to make it easier to read.

  1. There are one or two recent (February and March 2024) references that can be cited as they are relevant with this work and should be discussed versus the current results.

Thanks to your suggestions, we have added some of the literature and hope to better explain our results.

  1. In line with the findings of the study.

Thank you for your careful reading.

Finally, thank you again for your professional review work on our manuscript.

Reviewer 2 Report

Comments and Suggestions for Authors

The study utilized ONT sequencing technology to obtain the full-length transcriptome the muscle of Jiangquan black pigs at gestational days 60 and 75, as well as at postnatal day 240. They identified 9374 novel transcripts and screeded 5 candidate genes, TNNI2, MYOZ1, PDLIM3, TMOD4 and ACTA1, which may play key roles in the skeletal muscle growth and development of Jiangquan black pigs. The article is well-structured and the authors provide a clear and concise introduction to the study, a detailed methods section, and a clear and concise discussion of the results. However, there are a few areas where the article could be improved.

1. The Abstract is incomplete because some important information is missing. The authors should provide the information about samples, and the main results.

2. Line 28, replace “pork” with “pigs”

3. Line 29, revise “It’s” as “Its”

4. Line 41-45, please polish this sentence.

5.Line 52-58, this section should provide examples to support the previous point, such as the different expression study between different breeds, development stages, or individuals with extreme performance. What’s more, I suggest deleting the sentence “Differential gene……pig breeds”.

6. Line 71, insert “not only” after “will”.

7. Line 73, revise “promotes” as “promote”.

8. All figures suffer low resolution. It’s essential to improve the quality to facilitate comprehension for readers.

9. Figure1, label the age for A, B and C to make them more readable.

10. Line 90, this means number or length? If this means number, please check carefully and ensure the number is consistence in the text and Table1.

11. Figure 2A, Known plus novel don't equal all. Please check these numbers.

12. Line101, put a space between the sentences. The need to insert a space occurs in several places in the manuscript.

13. Line131-133, please rephrase this sentence.

14.Line134, “These target genes were mostly regulated by TMOD4.t1 and PDLIM3.t5”. I don’t know how the author infer this conclusion. Please explain and provide more information about target genes.

15. Line160-162, the authors wrote that “among which the gene TMOD4 and the gene TNNT3 may play important roles in the development of skeletal muscle in Jiangquan black pigs.” Why are these two genes? According to the information authors provided, I can't understand this conclusion.

16. Line180, in Go analysis, “term”, not “pathway”, is generally used.

17. The Discussion section seems to be insufficient in depth. The authors payed more attention to the several genes and discussed them detailly. However, the GO and KEGG pathway enrichment analyses, as well as alternative splicing were not further explained. It would benefit from a more comprehensive analysis of the study's findings in the context of existing literature, as well as a discussion of potential limitations and future research directions.

Author Response

Reviewer 2

Dear Reviewer,

Thank you very much for taking the time to review this manuscript. I really appreciate all your comments and suggestions! We use this feedback to improve the quality of the manuscript. The comments below are in italics. Our responses are in normal font and changes/additions to the manuscript are in highlighted text.

Thanks again!

  1. The Abstract is incomplete because some important information is missing. The authors should provide the information about samples, and the main results.

As you suggested, we have modified our summary and the changes will be shown in highlighted text.

  1. Line 28, replace “pork” with “pigs”

We sincerely thank you for your careful reading, and in accordance with the reviewers' comments, we have changed "pork" to "pigs".

  1. Line 29, revise “It’s” as “Its”

We apologise for our carelessness and thank you for the reminder.

  1. Line 41-45, please polish this sentence.

Thank you for pointing out that we did our best to improve the manuscript and that these changes do not affect the structure and framework of the paper. We have highlighted text in the revised manuscript.

  1. Line 52-58, this section should provide examples to support the previous point, such as the different expression study between different breeds, development stages, or individuals with extreme performance. What’s more, I suggest deleting the sentence “Differential gene……pig breeds”.

Thanks to your suggestion, we added more references to prove our point. We have removed the sentence 5. "Differential gene ......pig breeds" as you suggested.

  1. Line 71, insert “not only” after “will”.

Thank you very much for your suggestions, I have revised my manuscript.

  1. Line 73, revise “promotes” as “promote”.

Thank you very much for your suggestions, I have revised my manuscript.

  1. All figures suffer low resolution. It’s essential to improve the quality to facilitate comprehension for readers.

Thanks for pointing out that there may be word file compression of the image quality, I will attach my original image.

  1. Figure1, label the age for A, B and C to make them more readable.

Thanks for pointing this out, I've reworked the image and labelled it with the sample info.

  1. Line 90, this means number or length? If this means number, please check carefully and ensure the number is consistence in the text and Table1.

Thank you for your careful reading, these numbers are clean reading paragraph numbers, I double checked my text and made sure it was consistent with Table 1. I apologise for my carelessness.

  1. Figure 2A, Known plus novel don't equal all. Please check these numbers.

Thank you for your careful reading, as a gene will have more than one transcript, during our counting process, a Novel Transcript was found for that gene, so that gene was also noted as a Novel Gene, hence the situation where the number of Novel Genes and the number of Known Genes add up to be greater than all the genes.

  1. Line101, put a space between the sentences. The need to insert a space occurs in several places in the manuscript.

Thank you for your careful reading, I inserted spaces and checked all the text.

  1. Line131-133, please rephrase this sentence.

Thank you for your suggestions, and I have done my best to revise the language in the manuscript.

  1. Line134, “These target genes were mostly regulated by TMOD4.t1 and PDLIM3.t5”. I don’t know how the author infer this conclusion. Please explain and provide more information about target genes.

Thank you for your expert advice, as you pointed out, we can't get to that conclusion from the information in the manuscript, so I've revised the sentence in the hope that it seems more logical.

  1. Line160-162, the authors wrote that “among which the gene TMOD4 and the gene TNNT3 may play important roles in the development of skeletal muscle in Jiangquan black pigs.” Why are these two genes? According to the information authors provided, I can't understand this conclusion.

Thank you for your careful reading, I have revised my results based on the information mentioned in the manuscript.

  1. Line180, in Go analysis, “term”, not “pathway”, is generally used.

Thanks to your suggestion, I changed the word and made it consistent throughout the text

  1. The Discussion section seems to be insufficient in depth. The authors payed more attention to the several genes and discussed them detailly. However, the GO and KEGG pathway enrichment analyses, as well as alternative splicing were not further explained. It would benefit from a more comprehensive analysis of the study's findings in the context of existing literature, as well as a discussion of potential limitations and future research directions.

Thanks to your suggestion, I have added a discussion of pathways for Alternative Splicing and enrichment analysis in the Discussion section, at lines 334-355 of the article, which I show in the text using highlighted text.

Finally, thank you again for your professional review work on our manuscript.

Round 2

Reviewer 1 Report

Comments and Suggestions for Authors

The authors have addressed correctly all the points and have improved the manuscript. No further comments.

Author Response

Thank you for your valuable comments that made the revised manuscript more readable and scientifically sound.

Reviewer 2 Report

Comments and Suggestions for Authors The concerns that I proposed are all properly addressed. Now I think that this revised manuscript can be accepted for publication.

Author Response

(The authors gave the same response as above.)
